# Recurrent Vulvovaginal Candidiasis: An Immunological Perspective

**DOI:** 10.3390/microorganisms8020144

**Published:** 2020-01-21

**Authors:** Diletta Rosati, Mariolina Bruno, Martin Jaeger, Jaap ten Oever, Mihai G. Netea

**Affiliations:** 1Department of Internal Medicine and Radboud Center for Infectious Diseases, Radboud University Medical Center, 6525 GA Nijmegen, The Netherlands; diletta.rosati@radboudumc.nl (D.R.); mariolina.bruno@radboudumc.nl (M.B.); Martin.Jaeger@radboudumc.nl (M.J.); Jaap.tenOever@radboudumc.nl (J.t.O.); 2Department for Genomics & Immunoregulation, Life and Medical Sciences Institute (LIMES), University of Bonn, 53115 Bonn, Germany

**Keywords:** vulvovaginal candidiasis, recurrent vulvovaginal candidiasis, *Candida albicans*, host immune response, immunity, fungal infections

## Abstract

Vulvovaginal candidiasis (VVC) is a widespread vaginal infection primarily caused by *Candida albicans*. VVC affects up to 75% of women of childbearing age once in their life, and up to 9% of women in different populations experience more than three episodes per year, which is defined as recurrent vulvovaginal candidiasis (RVVC). RVVC results in diminished quality of life as well as increased associated healthcare costs. For a long time, VVC has been considered the outcome of inadequate host defenses against *Candida* colonization, as in the case of primary immunodeficiencies associated with persistent fungal infections and insufficient clearance. Intensive research in recent decades has led to a new hypothesis that points toward a local mucosal overreaction of the immune system rather than a defective host response to *Candida* colonization. This review provides an overview of the current understanding of the host immune response in VVC pathogenesis and suggests that a tightly regulated fungus–host–microbiota interplay might exert a protective role against recurrent *Candida* infections.

## 1. Definition and Clinical Perspective

Vulvovaginal candidiasis (VVC) is a debilitating pathological condition caused by *Candida* species, commonly characterized by vulvar itching, burning, pain while urinating, and vaginal discharge [1]. *Candida* is a dimorphic fungus from the phyla Ascomycota that inhabits the respiratory, gastrointestinal and genitourinary tracts of more than 30% of healthy individuals during their lifetime [2,3]. Alterations in the symbiotic interplay between the fungus and the mucosal ecosystem are associated with mild to moderate fungal dysbiosis, depending on the infected area as well as the patient’s health status. After anaerobic bacterial vaginosis, VVC is considered the second most common vaginal infection, affecting 75–80% of women at least once in their lifetime [1,4]. Despite the different pathogenesis, symptoms of fungal and bacterial vaginitis are often confused, thus resulting in women having an inaccurate diagnosis and reduced quality of life [5].

Up to 9% of women in various populations experience more than three or four episodes within one year, which is regarded as recurrent vulvovaginal candidiasis (RVVC) [6]. Worldwide prevalence and epidemiological data are rare and inaccurate because they are mostly carried out from self-reports and local general practitioner diagnosis. In this regard, Denning et al. systematically assessed epidemiological studies from 1985 to 2016 and, basing their study on the 6000 online surveys from five Western European countries and the United States by Foxman et al., documented a global annual prevalence of 3871 RVVC cases per 100,000 women, with the highest frequency (9%) in patients aged between 25 and 34 years old [6,7]. According to the Clinical Practice Guidelines, VVC can be treated with topical or oral antifungal formulations, among which azoles (e.g., miconazole, clotrimazole and fluconazole) are the most frequently prescribed therapeutics [8], although they do not prevent recurrent episodes after therapy cessation, necessitating antifungal prophylaxis [9]. RVVC does not correlate with mortality rates but the morbidity is dramatically increasing, and the costs associated with medical care rise accordingly. Hence, more effort needs to be made on the one hand to understand the immunopathogenesis and on the other hand to treat VVC patients efficiently and prevent recurrences.

In this review, we first provide a brief overview of the risk factors associated with increased susceptibility to VVC and then focus on RVVC immunology and pathogenesis. We hypothesize that RVVC might be due to a dysregulated immune system in response to *Candida* colonization rather than a defective host defense.

## 2. Risk Factors Associated with RVVC Susceptibility

Vulvovaginal candidiasis is considered a multifactorial disorder, where an imbalanced vaginal microbiota composition, host predisposing factors and genetics as well as *Candida* strains are likely to favor disease onset (Figure 1). The vaginal microbiome is commonly inhabited both by bacterial communities, mainly represented by the genus *Lactobacillus*, such as *L. iners* and *L. crispatus*, and yeast counterparts, commonly called the *mycobiome* [10,11]. *Candida* species are the most abundant fungal organisms of the vaginal mycobiome; hence, they can be causative agents of vaginal infections under some conditions [12,13,14]. *Lactobacilli* species are believed to favor a healthy vaginal microbiome both by acidifying the environment through anaerobic metabolism of glycogen to D-lactic acid and through hydrogen peroxide (H_2_O_2_) production, whose antimicrobial activity is likely to inhibit *Candida* invasion [15,16,17,18]. Several factors can alter the vaginal microbiota in patients with RVVC: firstly, changes in the H_2_O_2_-producing *Lactobacillus* community (e.g., *L. acidophilus*, *L. gasseri* and *L. vaginalis*), and secondly, a high estrogen condition (i.e., estrogen replacement therapy, luteal phase or pregnancy) as well as diversity in carbon sources, short-chain fatty acids or eicosanoids composition (Table 1) [19,20,21,22,23,24,25]. These conditions have been shown to disrupt the balance between tolerance and invasion, leading to *Candida* adherence to the mucosal epithelium, abnormal yeast growth and increased risk of contracting *Candida* infections [26,27].

In addition, a broad spectrum of host-related predisposing factors such as type-2 diabetes mellitus, immunosuppression regimens, antibiotics therapy, as well as behavioral factors such as use of contraceptives and intrauterine devices have been suggested to promote the onset of VVC [29,30,31]. However, since around 20–30% of VVC patients are healthy women without predisposing factors, it has also been suggested that inter-individual differences such as genetic background and ethnicity, as well as types of *Candida* strains and occurrence, might play a key role in idiopathic RVVC pathogenesis.

According to epidemiological data and multi-ethnic cohort studies, increased susceptibility to RVVC rates correlates with genetic polymorphisms as well as ethnicity. For instance, carriage of the single nucleotide polymorphism (SNP) in exon 1 codon 54 in the mannose-binding lectin 2 (*MBL2*) gene is reported to be more frequent in RVVC patients than healthy women, and this association has also been revealed in Brazilian, European, Chinese and Egyptian, but not Italian women [32,33,34,35,36]. In addition, clinical reports showed that African Americans are more prone to fungal infections in comparison to European or Hispanic women, thus suggesting a variation in frequency among ethnic groups [37,38].

Furthermore, it has emerged that the incidence of *Candida* infections is also species-related. Distribution and epidemiological studies carried out on cohorts in the United States, Europe and Australia identified *C. albicans* as the main occurring species, isolated in 75–90% of the positive cultures for VVC [39,40,41]. As far as the non-*albicans Candida* (NAC) infections are concerned, the highest frequency rates have been reported for *C. glabrata*—around 10–20% of cases, followed by *C. parapsilosis*, *C. tropicalis*, *C. krusei* and *C. africana* [42,43,44]. Given that species distribution and incidence vary among countries as well as the population studied, a significant increasing occurrence of NAC species in VVC patients has been revealed [45,46]. In particular, this has been reported in Tunisia, Nigeria, Middle Eastern countries and Asia, where *C. glabrata* is the most frequently isolated NAC (30–50%) [43,44,47,48,49]. In addition, it is believed that NAC species are more likely to favor recurrent infections in VVC patients [43,50], perhaps because of their refractoriness to the current azole drug treatment [31].

Interestingly, a correlation between genotype distributions of *C. albicans* and their severity has emerged from cohort studies in China, where high isolation rates of the dominant genotypes of *C. albicans* were more frequent in severe VVC compared to mild-to-moderate vaginal isolates [51]. Nevertheless, this correlation was not validated in Turkish women, suggesting therefore a different distribution pattern of *C. albicans* genotype between Chinese and non-Chinese VVC patients [52,53]. According to a microsatellite typing study in women affected by acute VVC, non-*albicans* species have been found to be potentially more severe than *C. albicans* species, the latter being mainly associated with a mild or moderate phenotype [54]. Although the aforementioned predisposing causes are believed to explain some cases of VVC worldwide, it remains to be elucidated what contributes to the shift from VVC to RVVC. Therefore, it may be hypothesized that an immunological dysregulation by the host might explain the majority of episodes, although it has yet to be clarified whether it may be due to a deficient or hypersensitive response of the host immune system.

## 3. Immunology: RVVC as an Immunodeficiency

Our knowledge about the immune response upon invasion of pathogens has advanced considerably during recent decades. In this regard, for more insights, comprehensive reviews have been published in the literature on this subject [55,56]. Briefly, once *C. albicans* impairs the first line defense, represented by epithelial cells, antimicrobial peptides and signaling molecules are released, thus leading to innate immune cell recruitment. Different fungal cell-wall components such as β-glucans and mannoproteins, commonly referred to as pathogen-associated molecular patterns (PAMPs), are sensed by several families of pattern recognition receptors (PRRs), primarily Toll-like receptors (TLRs) and C-type lectin receptors (CLRs) expressed on myeloid cells. Once recognized, *Candida* components are processed by phagocytic cells and, based on antigen specificity, T helper (Th) cells are differentiated in cellular subsets, leading ultimately to pathogen clearance [57].

Host genetic variants in PRRs as well as in signal transducers have long been thought to impair antifungal immune response in RVVC patients. For instance, a non-synonymous mutation in TLR2 (rs5743704, Pro631His) was reported to mediate an enhanced RVVC susceptibility [58]. Moreover, several cohort studies in different countries have investigated a SNP in the first exon of the *MBL2* gene, which is commonly associated with a defective immune response in various diseases [59]. However, no central role for *MBL2* polymorphism as the sole predisposing factor, either in protecting or in favoring RVVC onset, has been demonstrated, thus strengthening the concept that RVVC is a multifactorial disorder [35,60]. For its essential role in β-glucans recognition, Dectin-1 has been largely studied in antifungal immunity: knockdown of the receptor in mouse experiments resulted in higher fungal burden and a defective Th1/Th17 response, highlighting that antifungal mucosal immunity relies on host genetics [61,62]. In addition, it has been reported that a mutation in the human *CLEC7A* gene is associated with an impaired cytokine response and an increased susceptibility to recurrent mucosal or skin infections, but not oropharyngeal candidiasis (OPC) [63], rather than invasive candidiasis [64,65]. Hence, it has been long believed that VVC onset is the result of SNPs or other genetic alterations in key signaling proteins. Although the data suggest that the homozygous point mutation in Dectin-1 may be crucial for severe mucosal, but not systemic, infections, it is rare and limited to a small number of individuals—around 10–15% of Western women are heterozygous and less than 1% are homozygous for this polymorphism. Whether monoallelic variants that are commonly associated with a decreased immune response to fungal infection are responsible for RVVC pathogenesis remains to be elucidated.

In addition to genetic variations, a compromised antifungal host defense, such as that found in diabetes mellitus, is thought to increase the risk of vaginal fungal infections [66]. Indeed, cross-sectional studies in Brazil confirmed that *Candida* colonization rates are higher in diabetic women (18.8%) than in non-diabetic patients (11.8%), with the diabetic group showing more symptomatic (VVC + RVVC = 66.6%) than colonized (33.3%) patients [67]. Even though uncontrolled diabetes causes metabolic alterations that can predispose women to symptomatic vaginitis, only a minority of women are affected by RVVC, suggesting individual differences or additional factors may influence susceptibility to *Candida* infections.

Since a correlation between susceptibility to oral, gastrointestinal and mucocutaneous candidiasis is linked to T-cell deficiency, both human and animal studies have been carried out in recent decades to elucidate whether an impaired cell-mediated immunity (CMI) is also responsible for the lack of protection against VVC [68]. As Th1 and Th17 responses are known to be primarily involved in protection against candidiasis, clinical studies directed their attention toward a systemic immunosuppression state due to an impaired T-cell-mediated immune response. For instance, inherited genetic mutations in the Th17 pathway (e.g., STAT1, STAT3, STAT4) are known to lead to several diseases such as chronic mucocutaneous candidiasis (CMC), Hyper-IgE syndrome (HIES) and autoimmune polyendocrinopathy–candidiasis–ectodermal dystrophy (APECED). The result is an impaired IL-17 immune response, subsequent decreased neutrophil recruitment and hence a higher fungal load [69]. Interestingly, these individuals are highly prone to develop mucosal candidiasis such as recurrent OPC or onychomycosis, but not RVVC, despite some cases of vulva lesions having been described in both a Brazilian and an Iranian family [70,71,72]. Case-controlled studies in patients with acquired disorders of T-cell immunity, such as HIV/AIDS, revealed a distinct protective host defense mechanism against OPC and VVC. In fact, RVVC occurs equally among immunocompetent and immunocompromised women, while OPC is more frequent under immunocompromised states such as in HIV-positive individuals (>90%) [73,74,75]. In addition, experimental estrogen-dependent murine models reported an absence of systemic T-cell infiltration into the vaginal mucosa and normal levels of Th-1/Th-2 cytokines between RVVC patients and controls [76,77,78]. These results are in accordance with a cohort study by Talaei et al., who measured an altered T-lymphocyte proliferation in response to *Candida* in only 58% of RVVC patients compared to controls [79]. Since clinical observation revealed that mucosal sites are differentially susceptible to *Candida* infection and mice models of vaginal candidiasis failed to identify a protective role of Th-1 and Th-17 phenotypes, a local dysregulation in CMI rather than a systemic immune deficiency has been considered to be involved in RVVC susceptibility [75]. A murine model of vaginal candidiasis reported higher vaginal tissue expression and production of TGF-β1, thus suggesting its immunoregulatory activity might be responsible for a lack of CMI at the level of the vagina [76,80]. However, animal model studies failed to identify a protective role for resident vaginal T-cells upon *Candida* infection [76,81,82]. Moreover, evaluation of human *Candida*-specific antibodies (IgG and IgA) in serum and vaginal secretions both in humans and mice did not provide any significant evidence that humoral immunity (HI) mediates RVVC protection [83,84,85,86].

Overall, although our knowledge has advanced considerably during recent years, the role of cell-mediated immunity remains limited. While some sub-groups of VVC patients, such as those with CLEC7/dectin-1 deficiency, may suffer from RVVC, classical immune deficiencies characterized by defects in antifungal host defense (e.g., HIV-infected patients, patients with CMC) fail to show an increased susceptibility to VVC, which suggests that there are additional pathways for the pathogenesis of the disease.

## 4. Immunology: RVVC as an Autoinflammatory Disease

Given the unconfirmed protective role of T-cells in modulating the host response to vaginal candidiasis, the idea of a dysregulated innate immune response to *Candida* colonization has gained a foothold in recent years.

Even though animal models are indispensable for the study of RVVC immunopathogenesis, they are not completely able to reproduce the human condition: in fact, high exogenous estrogen administration, as well as a large *Candida* inoculum, are required to establish persistent infections in mice in which *Candida* is absent from the normal microbiota [87]. To shed light on the immune regulation of *Candida* infection, Fidel and colleagues challenged healthy women intravaginally with live *Candida* and found that the protection against infection is non-inflammatory; in contrast, symptomatic infection correlates with elevated vaginal polymorphonuclear neutrophil (PMN) infiltration and fungal burden, thus suggesting that symptoms are PMN-mediated [88]. In accordance, high levels of neutrophils have been reported in vaginal lavages of VVC patients [89]. Vaginal fluids from mice inoculated with *Candida* exhibited high levels of epithelial-derived S100A8 calcium-binding protein (also known as calgranulin A). S100A8 is an endogenous danger signal involved in the amplification of inflammation in autoimmunity as well as in response to infections [90,91]. Although alarmins are potent chemoattractants of PMNs in the vaginal lumen, they are sufficient but dispensable for initiating the acute neutrophil response upon interaction with small numbers of *Candida* in symptomatic individuals [92]. However, since activated phagocytes fail to reduce the fungal burden and vaginal epithelial cell anti-*Candida* activity is merely fungistatic [93], other mediators are thought to be involved in PMN dysfunction once recruited in the vaginal mucosa. Interestingly, the proteoglycan heparan sulfate (HS) has been recently indicated as a novel candidate involved in the lack of PMNs’ killing activity, placing them in an anergic state once recruited in the vaginal epithelium [94,95].

Transcriptomic analysis of vaginal tissue from a murine model of VVC demonstrated an increased expression of host genes involved in myeloid cell mobilization, phagocyte infiltration, as well as Th17 cytokine secretion (IL-22, IL-17A and IL-17F). It has also been demonstrated that NLRC4, IFN-γ, Nrh1 and miR-21 mediate NLRP3 inflammasome expression, subsequently leading to interleukin-1β (IL-1β) cleavage and infiltration of PMNs into vaginal tissue, as also reported in vaginal lavage fluid [96,97,98]. Indeed, RVVC patients carrying a polymorphism in the *NLRP3* gene had a higher production of IL-1β but low IL-1Ra levels, strengthening the central role of NLRP3 in favoring a persistent hyperinflammatory state in VVC [99]. Alteration of the vaginal microbiota, and diet-induced changes in composition as well as concentration (e.g., impaired tryptophan or short-chain fatty acid levels) have been shown to impact host immune response to *Candida* colonization by driving pathogenic NLRP3 inflammation in the absence of IL-22. A mouse model of VVC showed that stimulation of the aryl hydrocarbon receptor (AhR) on group 3 innate lymphoid cells (ILC3) by its ligand indole-3-aldehyde (3-IAld) promotes IL-22 expression, which in turn induces NLRC4 phosphorylation and subsequent NLRP3 restrained bioactivity via IL-1Ra production [100,101,102]. Disruption of the crucial IL-22/NLRC4/IL-1Ra axis has been addressed as being responsible for the lack of antifungal resistance, thus suggesting IL-22 deficiency might be a risk factor for RVVC susceptibility [103]. In addition to NLRP3, the inflammasome component NLPR6 has been found to trigger IL-18 production during the early stages of infection. Stimulation of the IL-22/NRLC4 axis by IL-18 production results in IL-18 release during a later phase, subsequently mounting a positive feedback loop mechanism. As deficiency in IL-18/IL-22 crosstalk is associated with impaired Th/Treg cell development and increased VVC susceptibility, IL-18 appears to act as a bridge between innate and adaptive immunity during vaginal candidiasis [104].

A dual role of IL-9 has been reported in gastrointestinal candidiasis [105]. Studies in mice showed that IL-9 is likely to modulate vaginal response to *Candida* in a time-dependent manner, initially favoring inflammation via NLRP3 activation and then tolerance to the fungus by stimulating IL-1Ra production, mast cell (MC) engagement and macrophage crawling [106,107]. Interestingly, as the IL-9/MC axis contributes to chronic allergic inflammation [108] and is associated in a sex-specific manner with IgE levels in cystic fibrosis [109], it may lead to the hypothesis that VVC is an allergic reaction to the ability of *Candida*-specific IgE and prostaglandin E2 (PGE_2_) to inhibit vaginal cell-mediated immune response [110]. These findings are consistent with a positive correlation between sensitization to atopic reactions and non-responders to RVVC maintenance treatment [111]. However, this phenomenon has been observed only in a minority of women, and further investigations are needed [112].

In accordance with transcriptomic results from the mouse model, a comprehensive genome-wide study in RVVC patients found that cell morphogenesis, adhesion and cellular metabolism pathways are dysregulated in RVVC patients. Furthermore, the combination with immunological data revealed that T-cell cytokines IL-17, IL-22 and IFN-γ were higher in individuals bearing a variant in the *SIGLEC15* (sialic acid-binding immunoglobulin-like lectin 15) gene, resulting then in a hyperinflammatory state in response to *Candida*. Given the fact that sialic structures are also expressed on the surface of *Candida*, *SIGLEC15* has been suggested as a novel susceptibility factor in RVVC [113]. Even though IL-23p19^−/−^, IL-17RA^−/−^ and IL-22^−/−^ mice intravaginally inoculated with *Candida* suggested little-to-no evidence of Th17 cytokines being primary effectors during acute inflammation [114], the role of IL-22 has been long investigated recently for its ability to provide antifungal resistance during infection at the mucosal sites by preventing epithelial cell damage [115]. Interestingly, a mouse model of VVC revealed that the role of the IL-17/IL-22 axis is independent from estrogen administration [116].

Overall, on the basis of mouse and human studies, a PMN-mediated immunopathology in symptomatic VVC/RVVC has been proposed: under certain vaginal environmental conditions, *Candida* switches from the commensal to opportunistic state by expressing a broad range of hypha-associated morphogenetic and virulence factors (nicely reviewed in [117])—mostly secreted aspartyl proteases (SAPs) and candidalysin. Inflammatory mediators such as alarmins and IL-1β produced by vaginal epithelial cells lead to PMN chemotaxis and recruitment to the vaginal mucosa. This event, in turn, induces elevated infiltration levels of the same mediators, establishing a positive feedback mechanism loop that results in a hyperinflammatory state [95]. Notably, the same pattern of inflammatory mediators has been reported in VVC patients, with little-to-no expression in asymptomatic colonized carriers and healthy women, providing evidence that NLRP3 is a hallmark of symptomatic immunopathology in humans [118]. Hence, VVC/RVVC might result from an overly aggressive innate reaction by PMNs, rather than an impaired adaptive immune response (Figure 2) [88,119]. Further studies are needed to elucidate the mechanisms that contribute to (R)VVC pathogenesis.

## 5. Perspectives and Future Directions

To date, data both from murine and human studies suggest that the immunological profile differs among RVVC patient subgroups: whereas deficiencies in the immune response to fungal infections characterize a small number of patients, the majority display a hyperinflammatory response towards *Candida* colonization and invasion, which might be due to the host’s genetic background. Of note, on the contrary to what one would expect if RVVC were an immunodeficiency, an acute response has been revealed in patients infected by very low fungal loads [120]. For these reasons, *Candida* burden alone cannot be considered predictive of disease. In this regard, a combination of genetic studies in RVVC patients and carefully matched controls combined with immunological data is needed to elucidate the role of the innate/adaptive immune response and decipher the factors involved in vaginal mucosal defense mechanisms upon *Candida* infection.

Even though antifungal agents are effective in patients, they provide static effects and do not prevent recurrences in roughly half of patients. Clarification of resistance mechanisms could improve the development of long-lasting and efficient drugs that target the inflammatory reaction. Adjuvant immunotherapy approaches have been developed over the years: humanized neutralizing antibodies against IL-9 [106], IL-1Ra [103] or molecular agents able to promote protection signaling to *Candida* infection, such as AhR agonists [104]. They could represent an option to ameliorate inflammation and alleviate symptoms in acute VVC patients, but future studies to demonstrate their efficacy are needed. Alongside immunotherapies, vaccination is a possible approach to boost the immune system and provide long-lasting memory of *Candida*. As an example, recombinant vaccines such as NVD-3, based on the virulence factor Alsp3 (agglutinin-like sequence proteins), and PEV-7 based on Sap2p [121,122] have progressed in the first trial steps. However, no vaccine is currently available in the clinical setting [123]. In addition, it has been proposed to use exogenous *Lactobacilli* administration—such as probiotics—as an adjuvant therapy, which might restore the microenvironmental balance and prevent VVC recurrences, since an abnormal vaginal microbiota is considered to favor reproductive tract infections. A recent study of vaginal microbiome transplantation in bacterial vaginosis may provide a template for such future studies [124]. Only a comprehensive approach to investigate all these potential treatments will be able to improve the outcome of this disease with high morbidity among women.

## Figures and Tables

**Figure 1 microorganisms-08-00144-f001:**
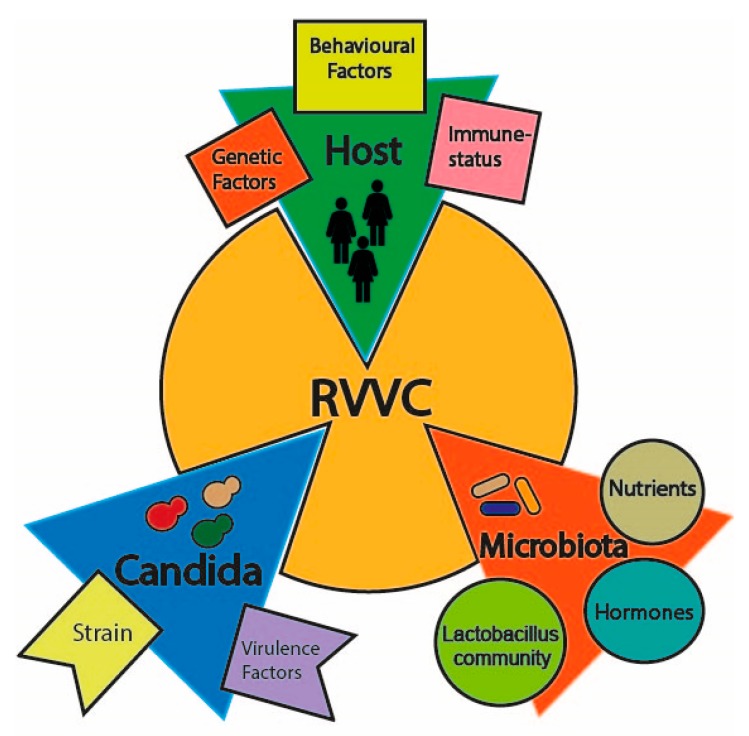
The factors contributing to recurrent vulvovaginal candidiasis (RVVC) onset.

**Figure 2 microorganisms-08-00144-f002:**
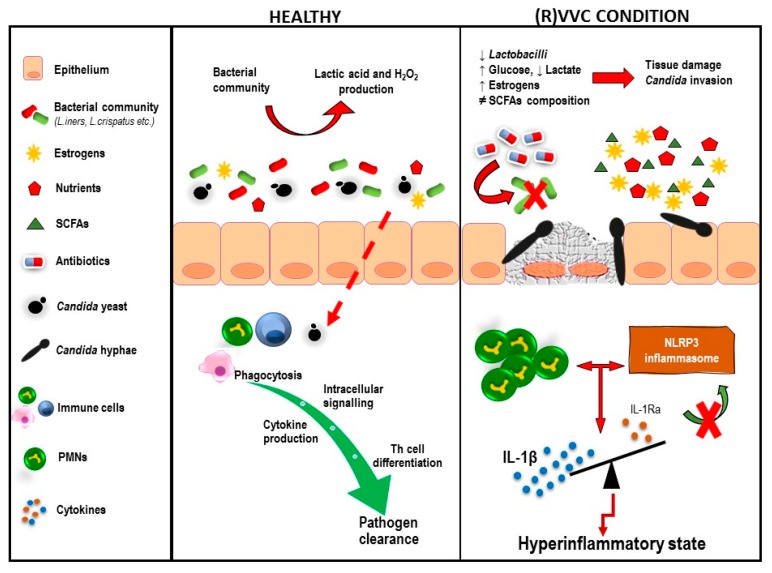
Environmental and immunological components that contribute to the host overreaction during RVVC.

**Table 1 microorganisms-08-00144-t001:** Summary of the microbiological factors that act in quorum sensing of vaginal microbiota with potential stimulatory or inhibitory effects on *Candida* growth/morphological switch.

Factors	Potential Effects on Candida Growth	Reference
*Lactobacilli* community	Inhibitory	[17,18,19]
Carbon sources:		
Glucose	Stimulatory	[24,25]
Lactate	Potentially inhibitory	[21]
Short-chain fatty acids *(SCFAs):*		
Acetate	Inhibitory	[21,22]
Butyrate	Inhibitory	[22]
Propionate	Inhibitory	[22]
Eicosanoids:		
Prostaglandin E2	Stimulatory	[23]
Thromboxane B2	Stimulatory	[23]
Estrogens concentrations	Stimulatory	[20]
Low *pH* (pH: 4–4.5)	Potentially inhibitory	[28]

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
