# Peer review of "Recurrent Vulvovaginal Candidiasis: An Immunological Perspective"

_microorganisms, 2020, doi:10.3390/microorganisms8020144_

Round 1

Reviewer 1 Report

This is a interesting review that summarizes the latest studies on the understanding of the host immune response in VVC pathogenesis. I recommend publication of this manuscript after some minor changes as below. > The authors concluded that tightly regulated fungus-host-microbiota interplay might exert a protective role against recurrent  of Candida infections. So, The authors should include a table with the summary of microbiological factors that act in quorum sensing of vaginal microbiota with potential stimulatory or inhibitory effects on Candida sp. growth

Author Response

Dear Reviewer,

We thank you for the nice comments and useful suggestion.

As suggested, we are interested in including a table with the summary of microbiological factors that act in quorum sensing of vaginal microbiota with potential stimulatory or inhibitory effects on Candida growth/morphological switch.

Here you find attached the proposed table and we strongly agree it could improve the manuscript. We hope you will find it suitable. If so, the references will be part of the text (so in the right coloumn there will be the numbers according to the bibliography).

We thank again for the careful reading and your valuable input.

We are looking forward to hearing from you,

Best regards,

Diletta Rosati

Reviewer 2 Report

This review article describes in concise and comprehensive manner the recent advances in research regarding recurrent vulvovaginal candidiasis, a disease affecting up to 9% of the women. Though other recent review articles have focused on vulvovaginal candidiasis, here the authors specifically discuss the importance of the host immune response for the development of RVVC. By providing multiple references, the authors manage to convince the reader that it appears that overreaction of the host immunity rather than immunodeficiency (as with other Candida spp. infections) is the key factor contributing to the recurrent candidiasis episodes in the affected women. The athours also discuss gaps in the field and propose directions for further research, thus increasing the strength of this review. The authors have done an excellent job in producing a well-written and rounded manuscript. However, few typos or not fully comprehensive sentences were noted (lines 58, 105, 139, 148, 151, 164). On a personal note, I am not very found of Fig 1. Although it may be too complex, I recommend to add a figure illustrating the components identified to contribute to the host response overreaction to RVVC. All in all, an outstanding review!

Author Response

Dear Reviewer,

We thank you for the nice comments and useful suggestions.

As suggested, we added a new figure illustrating the components that contribute to the host overreaction during RVVC and we believe it will improve the manuscript.

Next to that, we made modifications in the sentences mentioned and added citations to make the manuscript more clear to the reader (any revision made in the manuscript is mentioned in the cover letter and highlighted in the new version of the manuscript)

We thank you again  for the careful reading and your valuable input.

We are looking forward to hearing from you,

Best regards,

Diletta Rosati
